

# Effects of grazing intensity on forage nutritive value of dominant grass species in Borana rangelands of Southern Ethiopia

Yeneayehu Fenetahun[1,2], Yuan You[2], Tihunie Fentahun[3], Xu Xinwen[2] and Wang Yong-dong[2]

[1] University of China Academy of Sciences, Beijing, China
[2] National Engineering Technology Research Center for Desert-Oasis Ecological Construction, Xinjiang Institute of Ecology and Geography, Chinese Academy of Sciences, Urumqi, China
[3] College of Natural and Computational Sciences, Mekdela Amba University, Tulu Awuliya, Ethiopia

## ABSTRACT

**Background**. Forage nutritive value analysis is an essential indicator of rangeland status regarding degradation and livestock nutrient demand. Thus, it is used to maintain healthy and sustainable rangelands that can provide the livestock with sufficient quantity and quality of forage. This study is conducted with the aim of investigating the effects of grazing intensity combined with seasonal variation on the nutritive values of dominant grass species in the Teltele rangeland.

**Methods**. The studied area is classified into no-grazed, moderately grazed, and overgrazed plots based on the estimated potential carrying capacity. Sampling data is collected during both rainy and dry seasons. The collected forage samples are analyzed for concentrations of crude protein (CP), acid detergent organic fiber (ADF), neutral detergent fiber (NDF), acid detergent lignin (ADL), ash, dry matter digestibility (DMD), potential dry matter intake (DMI), and relative feed/forage value (RFV).

**Results**. The results show significant ($P < 0.05$) effects of both grazing intensity and season to grazing intensity interactions on all forage nutrient content concentrations across all grass species both within and between treatments. The recorded CP concentrations of all grass species are high in the overgrazed site and low at the no-grazed site, while the fiber concentration is high in NG and low in OG. RFV data also varies greatly, with high value recorded in OG in the rainy season and low value found in NG mainly during the dry season. As a result, it is recommended that moderate grazing should be practiced on the study site to maintain the quality and quantity of forage and to manage it in a sustainable manner.

# INTRODUCTION

Rangelands are the primary and cheapest source of forage for livestock (*Ismail et al., 2014*). In most countries, including Ethiopia, the livestock industry largely relies on natural rangelands, such as the Teltele rangeland (*Adnew et al., 2018*). Livestock depending on such natural rangelands face highly fluctuating nutritive value of forage grass species, although

Corresponding author
Wang Yong-dong,
wangyd@ms.xjb.ac.cn

there is a wide array of grass species (*Gelayenew et al., 2016*; *Newman, Lambert & Muir, 2009*; *Vendramini, 2010*). The nutritive value of rangeland forages varies due to influencing factors like grazing intensity, soil type, water availability, maturity/stage of development, part of the plant (leaf *vs.* stem), season (rainy *vs.* dry), environmental factors (moisture and temperature), altitude and management practice (*Amiri & Mohamed, 2012*; *Henkin et al., 2011*; *Jank et al., 2014*; *Kaplan et al., 2014*; *Adesogan et al., 2011*). The nutritive value of rangeland forage is often evaluated to estimate the carrying capacity of the rangeland and assess animal performance (*Godari, Ghiyasi & Poor, 2013*). The selection of grass species for forage depends on the acceptability nature of grass, which is linked to the flavor (like smell, taste, and texture) and nutritive value of the forage (*Estell et al., 2014*). However, based on the density of acceptable forage species, it is impossible to estimate the nutrition quality of the forage in the grazing area (*Samuels et al., 2015*).

The productivity and health of grazing livestock mainly depend on the nutrition they obtain from grass species, including protein, fiber, and mineral elements (*Brisibe et al., 2009*; *Massey, Ennos & Hartley, 2007*). Therefore, key aspects to consider when evaluating forages include protein, fiber and mineral nutrient concentrations (*Juárez et al., 2013*). In the Teltele rangeland, the livestock population increases overwhelmingly and causes overgrazing. Year-round grazing without any rest later results in significant changes in both the productivity and forage nutritive value of the grass species (*Selemani et al., 2013*). If grazing intensity (GI) increases, there will be a decrease in neutral detergent fiber (NDF), acid detergent fiber (ADF) and lignin, and an increase in crude protein (CP) and dry matter digestibility (DMD) (*Cline et al., 2009*; *Smart et al., 2010*). *Derner (2009)* and *Njidda, Olatunji & Raji (2012)* reported that with increased stocking rate or grazing pressure, there is a decline in animal performance. The forage nutritional value shows a linear decrease in N and CP and a linear increase in NDF when it switches from rainy to dry seasons, since digestibility has decreased due to the declining leaf-to-steam ratio caused by high temperature (*Kirch et al. 2007*).

In the Teltele rangeland, local pastoralists share the communal grazing areas for grazing livestock year-round without rest. However, in the communal grazing areas, livestock overgrazes palatable grass species and causes rangeland degradation (*Asmare et al., 2017*). Changes in the forage nutritive value of communal rangeland areas have become a focus for many academicians when identifying the linkage between grazing intensity and forage nutritive value (*Schut, Gherardi & Wood, 2010*). Understanding these effects and managing accordingly are crucial for establishing proper grazing systems (*Xiajie et al., 2018*). And knowing the spatial and temporal changes in rangeland forage quality is essential for livestock farmers (*Wubetie et al., 2018*). Thus, estimating the influence of GI on forage quality is critical to updating knowledge for maintaining sustainable management of grasslands in Teltele. But, to date, although there are a number of studies on arid and semi-arid rangelands around the world, there is no documented study data about the impact of GI on forage nutritive value of dominant grass species in the Teltele rangeland. This is one of the major research gaps that need to be addressed for achieving substantial rangeland management through balancing grazing capacity and livestock performance.

To restore rangelands through evaluating the impact of GI on forage nutritive value, spatial methods comparable across all species within the studied site are needed to be adopted to obtain clear and measured data. This study aims to achieve the following objectives: (1) to evaluate the effect of grazing intensity on forage nutritive value of dominant grass species, and (2) to compare the nutritive values of dominant grass species during dry and rainy seasons under different grazing intensities. Accordingly, to assess and propose a solution for controlling GI within an appropriate range, the following question is put forward: How and to what extent does GI in combination with climate (seasonal) variation affect grass species productivity and nutritive value in the Teltele rangeland? Simply stated, null hypotheses proposed by this study are: (1) Variation of GI across grazing lands does not pose a significant impact on either forage nutritive value or sustainable rangeland management; (2) The primary productivity, GI, and livestock productivity are similar in both rainy and dry seasons.

## MATERIALS AND METHODS

### Site selection

Both site selection and data collection are done following the same procedures used previously by *Fenetahun et al. (2020)* in the study "Dynamics of forage and land cover changes in Teltele district of Borana rangelands, southern Ethiopia: using geospatial and field survey data". The study is conducted in the Teltele semi-arid rangeland in the Borana zone of Southern Ethiopia, by selecting areas that are no-grazed (NG), moderately grazed (MG), and overgrazed (OG) as a treatment using the calculated carrying capacity of those areas for two consecutive seasons in 2019 (Fig. 1) (*Fenetahun et al., 2020*). The study site is located between 04°56′23″N latitude and 37°41′51″E longitude (*Fenetahun et al., 2020*; *Dalle, Maass & Isselstein, 2015*), and it is selected because it is one of the most arid parts of Borana zone and, therefore, the pastoral communities of this region are the most vulnerable to the rangeland degradation as a result of overgrazing (*Fenetahun et al., 2020*). The area is located 666 km south of Addis Ababa, the capital city of Ethiopia (*Fenetahun et al., 2020*). The elevation is about 496–1,500 m with a maximum elevation of 2,059 m above sea level (*Fenetahun et al., 2020*). Rainfall is bimodal with the main (60%) rainy season occurring between March and May, while the short (27%) rainy season occurs from September to November (*Dalle, Maass & Isselstein, 2015*; *Fenetahun et al., 2020*) (Fig. 2). The two intervening dry seasons are from June to August and December to February, when forage resources are scarce (*Fenetahun et al., 2021*; *Angassa, 2014*). The mean annual rainfall recorded over the past 12 years (2008–2019) is 450–700 mm (NMA, 2019, and *Gemedo, 2020*), while the mean annual temperature varies from 28 to 33 °C with little seasonal variation (*Fenetahun et al., 2020*). The annual potential evapotranspiration of the area is 700–3,000 mm (*Billi, Alemu & Ciampalini, 2015*). The main soil type in the study area, composed of 53% sandy, 30% clay, and 17% silt, is mainly used to support the growth of grass species for grazing (*Fenetahun et al., 2020*; *Fenetahun et al., 2021*; *Coppock, 1994*; *Gemedo, 2020*). The rangeland vegetation is mainly dominated by encroaching woody species and those that are frequently thinned out, including *Senegalia mellifera,*
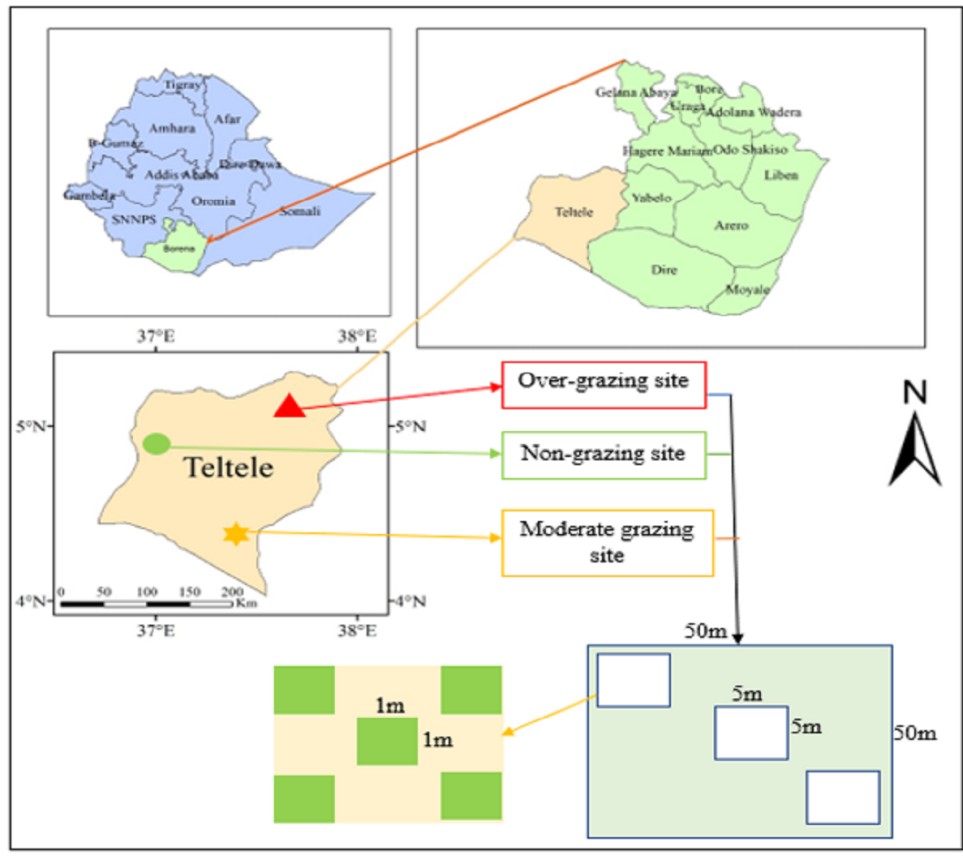

**Figure 1** Location map of the study area and sampling plot layout.

*Vachellia reficiens, and Vachellia oerfota* (*Coppock, 1994*; *Gemedo, Maass & Isselstein, 2005*; *Fenetahun et al., 2020*). The 2015 national census reported a total population of 70,501 for this woreda, including 36,246 men and 34,255 women, and 4,874 (6.91%) urban dwellers. At that time, cattle, goats, sheep, camel, mule, donkey, and horse were the main species of livestock species reared in the area (*Fenetahun et al., 2020*). Furthermore, according to data reported by the zone livestock office, the estimated total number of herds for all species is 201,148, and the proportions of each species found in the study district are: cattle 92,000 (45.7%), goats 58,139 (28.9%), sheep 17,210 (8.6%), camels 15,305 (7.6%), horses 8,000 (4.9%), mules 3,494 (1.7%), and donkeys 7,000 (3.5%). For the OG site, all species graze year-round without rest, as pastoralist migration from one area to another is highly limited by government policies, which is a major cause of overgrazing and greatly impacted the nutritive value of grass species in the Borana rangelands.

## Experimental design and sample collection

Data is collected using the same procedures described by *Fenetahun et al. (2020)*, as the same site is studied. We select the site using three treatments: a no-grazed (NG) site (as a control), a moderately grazed site and an overgrazed site (used to examine the effect

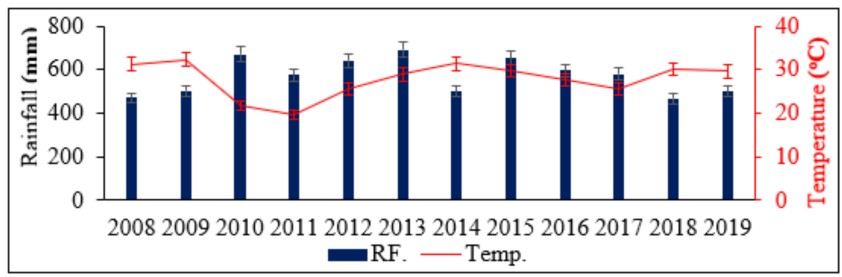

**Figure 2   Mean annual rainfall (RF) and temperature (Temp) from 2008–2019 in the Teltele rangeland site.**

of grazing intensity) based on grazing intensity gradient (*Fenetahun et al., 2021*) and the current carrying capacity potential (*Fenetahun et al., 2020*; *Fenetahun et al., 2021*). Inside the NG site, livestock has been abandoned, so the site is used as a reference to compare the forage nutritive value with other areas and to evaluate the impact of GI. The livestock species found in all grazing treatments are the same and the only variables are density and GI. Once the forage yield and utilization rate are determined, the carrying capacity (CC) can be calculated. The information can be used in two alternative ways: (a) to determine stocking rate or the number of heads of animals a system can carry in terms of the total livestock unit (TLU ha$^{-1}$ year$^{-1}$), or (b) to determine how much area a specific herd can graze in the system (ha TLU$^{-1}$ year$^{-1}$) (*FAO, 1988*). Similar to the CC calculation, 30% consumable rate is applied on the potential yield to calculate the stocking rate using the following formula:

$$\text{Stocking rate for the year (TLUha}^{-1}\text{ year}^{-1}) = \text{TLU/total grazing area} \qquad (1)$$

The treatment sites for sample collection involves a stocking rate of NG ($\sim$0 TLU ha$^{-1}$Y$^{-1}$), MG (2 TLU ha$^{-1}$ Y$^{-1}$), and OG (4 TLUha$^{-1}$ Y$^{-1}$ and above) based on the current forage biomass yield and carrying capacity of rangeland calculated by applying the approach proposed by *Fenetahun et al. (2020)* and through physical field observation. The treatments with different GI are selected and investigated at 2 km intervals (*Fenetahun et al., 2021*). The selected rangeland sampling areas of these three GI sites are 100 ha for each (in total one NG + one MG + one OG sites = 300 ha) (*Fenetahun et al., 2021*). The sites are selected from a homogeneous area and have similar geographical conditions like slope, elevation, and soil types (*Fenetahun et al., 2021*). The grazing treatments and sample collection are implemented both during the dry season (from December 2018 to February 2019) and the rainy season (from March to May 2019) at the time where grass species can be easily observed and peak biomass is recorded to evaluate the interactional effect of seasonal variation combined with GI, with three replications for each season (*EMA, 2019*; *Fenetahun et al., 2021*). Then, after establishing a 5 km transect both in the NG and grazing sites, five 50 m × 50 m plots at 500 m intervals are established, resulting in a total of 15 plots (three treatments with five plots each). Then, in each plot, three 5 m × 5 m subplots are randomly assigned as pseudo-replicates out of the total of 45 subplots in grazing and

non-grazing treatments (*Fenetahun et al., 2020*; *Fenetahun et al., 2021*). Finally, five 1 m × 1 m quadrats in each subplot, with a total of 225 quadrats per season (450 quadrats across the two sampling seasons) are assigned by randomly casting them back side to minimize any bias resulting from selective placement in each subplot for the collection of samples of grass species over two consecutive seasons (Fig. 1) (*Fenetahun et al., 2020*; *Fenetahun et al., 2021*). The total sampling of 5 plots × 3 sub-plots × 5 quadrats × 2 seasons × 3 replications = 450 for each treatment site is conducted (*Fenetahun et al., 2020*; *Fenetahun et al., 2021*). Moreover, the same sample collection techniques and treatment sites are adopted during the dry and rainy seasons (*Fenetahun et al., 2020*; *Fenetahun et al., 2021*). In each sampling unit, we record the dominant grass species and abundance for each grass species (*Fenetahun et al., 2020*; *Fenetahun et al., 2021*). All the ground grass samples are obtained by using a cutter and each grass species is stored separately in a paper bag (*Fenetahun et al., 2020*; *Fenetahun et al., 2021*). The fresh weight of the collected grass samples is measured in the field with a scale (*Fenetahun et al., 2020*; *Fenetahun et al., 2021*). Then, the samples are oven-dried for 48 h at 55 °C to value the biomass. The sub samples are used to calculate the dry weight of forage mass and estimate forage nutritive value as described below (*Fenetahun et al., 2020*; *Fenetahun et al., 2021*). The dried samples are first measured and grounded to pass a one mm screen for further analysis at a lab of College of Agriculture and Environmental Sciences, Bahir Dar University, Ethiopia. The evaluated forage comprising five dominant grass species (*Chloris roxburghiana*, *Cenchrus ciliaris*, *Chrysopogon aucheri*, *Aristida kenyensis* and *Digitaria milanjiana*) is selected and sampled based on relative abundances (≥40%) and pastoralists' experiences and preferences for certain grass species (*Habtamu et al., 2013*). Grass species are identified in the field by using identification keys, plates, the book *Flora of Ethiopia*, and the National Herbarium of Addis Ababa University (*Dalle, Maass & Isselstein, 2015*). The specific assessment for detailed acceptability values of dominant grass species and soil physicochemical properties is given by a study carried out on the same site and by the same author (*Yeneayehu, Xu & Wang, 2020*).

## Forage nutritive value analysis

Forage samples are analyzed for multiple quality factors on a dry mass basis, with a crude protein (CP), neutral detergent fiber (NDF), acid detergent fiber (ADF), acid detergent lignin (ADL), ash, dry matter digestibility (DMD), potential dry matter intake (DMI), and relative feed/forage value (RFV) following standard procedures as described below (Table 1). The calculated results of CP, NDF, ADF, and ADL are presented using g/kg as unit and ash, DMD, RFV and DMI are expressed in the form of percentage (%). Based on the obtained results of NDF, ADF, DMD and DMI, the forage nutritive quality of the grass species can be estimated and ranked by adopting the following formula (*Evitayani et al., 2004*; *Fazel, Abdul & Mohamed, 2012*). DMI, calculated as a percentage, estimates the relative amount of forage an animal will eat when only forage is fed (*Undersander, Mertens & Thiex, 1993*).

$$\%DMI = 120/\%NDF \tag{2}$$

**Table 1   Standard procedures and methods used to analyses forage nutritive value.**

| Major forage nutrition compositions | Analyses procedures and methods used | Reference |
|---|---|---|
| Crude protein (CP) | AOAC (1995) | Zhai et al. (2018) |
| Acid detergent fiber (ADF) | Acid detergent solution | *Van Soest, Robertson & Lewis (2015)* |
| Neutral detergent fiber (NDF) | Neutral washing liquid | *Van Soest, Robertson & Lewis (2015)* |
| Acid detergent lignin (ADL) | ANKOM 200 Fiber Analyzed | *Van Soest, Robertson & Lewis (2015)* |
| Ash contents | AOAC (1990) | Zhai et al. (2018) |
| Relative feed/forage value (RFV) | RFV= (%DMD × %DMI) ÷ 1.29 Where 1.29 = the expected digestible dry matter intake as % of body weight; DMD = 88.9 − (ADF% × 0.779), DMI = 120/% NDF. | *Newman, Lambert & Muir (2009)*; *Schacht et al. (2010)* |

Notes.
DMD, dry matter digestibility; DMI, dry matter intake.

$$RFV = \frac{DMD(\%) \times DMI(\%)}{1.29} \qquad (3)$$

RFV = Relative forage value of forage species predicted by NDF and ADF.

## Statistical analysis

Forage nutritive value data is analyzed using SPSS Version 22 with grazing treatment and season as well as their interactions as fixed factors, and plot considered as a random factor. Plot is treated as a repeated measure. There are 450 sample observations (5 plots × 3 sub-plots × 5 quadrats × 2 seasons × 3 replications) in each GI site for each forage variable (CP, ADF, ADL, DMD, DMI, NDF). Analysis for repeated measures concerning forage nutritive values is performed using a mixed model (Proc Mixed), including GI (NG, MG and OG), season (dry and wet), and their interactions as repeated effects. A two-way analysis of variance (ANOVAs) followed by a Duncan's multiple range test is performed to test significant differences ($P < 0.05$) between NG, MG, and OG treatments within and between each season. A simple linear regression analysis is conducted to examine the relationship between GI and various variables (from forage nutritive value response ratio to grazing in each season). A principal component analysis (PCA) is carried out to examine the relationship between forage nutritive values of the species based on the experimental results.

## RESULTS

### Effects of grazing intensity on forage nutritive value

The relative abundances of the selected dominant grass species across all grazing intensities and seasons are presented in Fig. 3. Based on the recorded data, *C. roxburghiana*, *C. ciliaris* and *C. aucheri* are >60%, *A. kenyensis* is ≥50%, and *D. milanjiana* is ≥40% in abundance both during the rainy and dry seasons. *C. aucheri* is the most abundant grass species in Teltele rangeland, followed by *C. ciliaris* and *C. roxburghiana*. The effects of GI on forage nutritive value varies significantly ($P < 0.05$) in terms of CP, ADF, NDF, ADL, ash, and

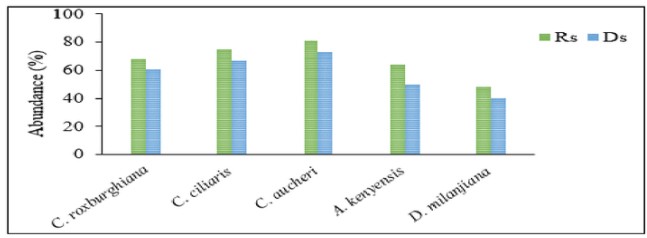

**Figure 3** Relative abundance of dominant grass species in the Teltele rangeland. Rs, rainy season, Ds, dry season.

**Table 2   Effects of grazing intensity on forage nutritive value of each grass species.**

| GI | Grass species | Forage nutrient compositions (%) | | | | | | | |
|----|--------------|------|------|------|------|------|------|------|------|
| | | CP | ADF | NDF | ADL | Ash | DMD | DMI | RFV |
| NG | *C. roxburghiana* | 2.4[a] | 42.7[aB] | 60.8[aB] | 29.1[aA] | 9.9[aaA] | 56.1[a] | 2.0[aaA] | 86.9[aA] |
| | *C. ciliar* | 7.7[bE] | 50.2[b] | 74.3[b] | 40.2[b] | 13.6[bB] | 49.8[b] | 1.6[bbB] | 61.8[b] |
| | *C. aucheri* | 5.5[cC] | 46.9[cA] | 71.7[c] | 39.4[b] | 12.1[cC] | 52.4[cA] | 1.7[dbB] | 69.0[c] |
| | *A. kenyensis* | 1.6[d] | 43.0[d] | 63.1[dC] | 32.2[cB] | 10.2[daA] | 55.4[aB] | 1.9[caC] | 81.6[d] |
| | *D. milanjiana* | 3.9[e] | 55.1[e] | 76.9[e] | 42.8[d] | 11.9[cC] | 46.0[e] | 1.6[bbB] | 57.0[e] |
| MG | *C. roxburghiana* | 4.1[cc] | 39.3[c] | 58.6[c] | 25.3[c] | 10.8[cA] | 58.3[cC] | 2.0[aaA] | 90.4[c] |
| | *C. ciliari* | 11.8[dD] | 48.3[d] | 70.1[d] | 33.1[d] | 14.8[d] | 51.3[d] | 1.7[bbB] | 67.6[d] |
| | *C. aucheri* | 8.4[e] | 41.1[ee] | 67.2[eA] | 30.5[e] | 13.5[bB] | 56.9[e] | 1.8[dC] | 79.4[e] |
| | *A. kenyensis* | 3.5[c] | 40.1[ee] | 59.6[c] | 28.7[bA] | 11.6[bbC] | 61.6[b] | 2.0[caA] | 89.5[b] |
| | *D. milanjiana* | 5.7[bA] | 51.8[b] | 72.5[b] | 36.5[a] | 12.5[abC] | 48.5[a] | 1.7[dbB] | 63.9[a] |
| OG | *C. roxburghiana* | 5.9[eA] | 36.5[e] | 56.1[e] | 23.1[e] | 11.8[b] | 60.0[e] | 2.1[aa] | 97.6[eb] |
| | *C. ciliari* | 15.2[c] | 43.6[aB] | 62.9[cC] | 30.7[c] | 15.0[a] | 54.9[aB] | 1.9[ebC] | 80.9[c] |
| | *C. aucheri* | 11.9[dD] | 38.3[dd] | 59.8[bB] | 26.9[d] | 14.2[d] | 59.0[ddC] | 1.9[bbC] | 86.9[dA] |
| | *A. kenyensis* | 4.9[bcC] | 37.8[cd] | 57.1[e] | 22.1[a] | 12.3[bC] | 59.4[cdC] | 2.1[ba] | 96.7[ab] |
| | *D. milanjiana* | 7.7[aE] | 46.4[aA] | 67.3[aA] | 32.5[bB] | 13.9[eB] | 52.8[bA] | 1.8[dbC] | 73.7[b] |

**Notes.**

Values in columns with different lowercase letters (a, b–etc.) are significantly different ($p < 0.05$) and values with the same second double lowercase letters under some treatment (aa, ba, cc, bc—etc.) and values with both lowercase and uppercase letters across the treatment (aB, –etc.) are indicated not significant difference ($p > 0.05$). GI, grazing intensity; NG, non- grazing; MG, moderate grazing; OG, over grazing; CP, crude protein; ADF, acid detergent fiber; NDF, neutral detergent fiber; ADL, acid detergent lignin; DMD, dry matter digestibility; DMI, dry matter intake; RFV, relative feed/forage value.

RFV contents both within and between each species (Table 2). For all grass species, the concentrations of CP and ash increase when GI rises, while the concentrations of ADF, NDF, and ADL decrease when GI drops. For the dominant grass species, the effect of GI on the forage nutritive value is described using linear regression analysis (Fig. 4).

According to the regression analysis results, the RFV of forage shows significant differences ($P > 0.05$) under different GIs. It presents a decreasing pattern if the concentrations of ADF, NDF, and ADL value are increasing, yet shows an increasing pattern when the concentrations of CP and ash value increase (Fig. 4). The RFV value could be used to estimate the forage quality. The higher the RFV value is, the higher the quality is. *C. roxburghiana*, A. *kenyensis,* and *C. aucheri* are grass species with high RFV (forage quality) based on our research data.

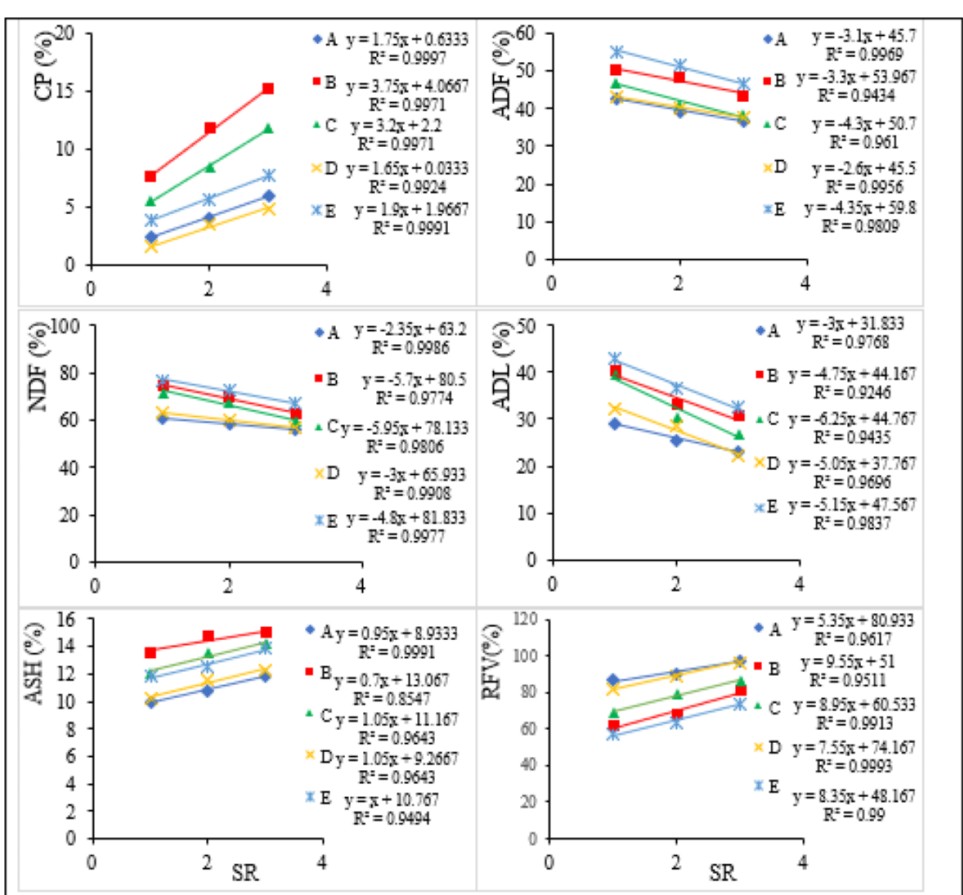

**Figure 4** **Relationship between stocking rate (SR) and forage nutritive values of each grass species.** A, *C. roxburghiana*; B, *C. ciliary*; C, *C. aucheri*; D, *A. kenyensis* and E, *D. milanjiana*.

In the NG site, the concentration of CP for all grass species is below the minimum requirements for beef cattle (7%) with the exception of *C. ciliar (7.7%)* and for small ruminants (9%) (*Gemedo, 2020*; *Habtamu et al., 2013*). Across all GIs, the concentration of CP for *C. roxburghiana, A. kenyensis* and *D. milanjiana* is below the minimum requirements for most grazing animals. This is because the grass maturity at the NG site increases and results in a lower leaf-to-stem ratio, leading to an increase in ADF, NDF and ADL, and a decrease in CP. At the NG site, the forage quality decreases as compared with both the MG and OG, due to decreasing digestibility and increasingly maturing proteins.

On the other hand, with an increasing GI, the rate of new growth declines, and the consumption of less desirable materials, such as the remaining part of mature forages from the previous growing season, results in a decreasing CP concentration. The concentrations of ADF, NDF, and ADL are highest for *D. milanjiana*, followed by *C. ciliary,* and lowest for *C. roxburghiana,* followed by *A. kenyensis*. The concentration shows a decreasing pattern when it switches from NG to OG and presents a significant difference ($P < 0.05$) both within and between species and across GIs. High ash content is recorded for *C. ciliary* and low for *C. roxburghiana*, showing a significant variation ($P < 0.05$) both within and

between species and across GIs, and increases at the OG site followed by the MG and the NG site respectively.

## The interaction effects of grazing and season on forage nutritive value

The nutritional composition of the forage grasses is significantly different ($p < 0.05$) among and within species due to the interactive effects of season and GI rate (Table 3). Therefore, the grazing season affects not only the biomass production of rangelands but also the nutritive value of existing forage grass species in the grazing site. The result indicates that the concentrations of CP, ash, DMD, and DMI content increase during the rainy season compared with those in the dry season. Across all grass species, the highest values of CP, ash, DMD, and DMI content are recorded at the OG site during the rainy season, whereas the lowest values are recorded at the NG site during the dry season.

The CP content varies from 1.3% (*A. kenyensis*) to 11.4% (*C. ciliari*) during the dry season and from 6.9% (*A. kenyensis*) to 18.9% (*C. ciliari*) during the rainy season. The ash content varies from 8.1% (*C. roxburghiana)* to 14.8% (*C. ciliari*) during the dry season and from 13.5% (*C. aucheri*) to 21.9% (*D. milanjiana*) during the rainy season, indicating that the concentrations of CP and ash are higher during the rainy season than in the dry season. The fiber constituents (*i.e.,* ADF, NDF, and ADL) of the forage grass species increase during the dry season compared with the values in the rainy season. The highest values of ADF, NDF, and ADL are recorded at the NG site during the dry season, whereas the lowest values are recorded at the OG site during the rainy season (Table 3). ADF, NDF and ADL vary from 29.5% (*C. roxburghiana*) to 47.1% (*D. milanjiana*), from 37.1% (*A. kenyensis*) to 60% (*D. milanjiana*), and from 13.1% (*A. kenyensis*) to 33.7% (*D. milanjiana*) during the rainy season respectively, and increase in the dry season: from 38.7% (*C. roxburghiana*) to 59.9% (*D. milanjiana*), from 59.3% (*C. roxburghiana*) to 88.5% (*C. ciliary*), and from 26.6% (*A. kenyensis*) to 53.1% (*C. ciliary*) respectively. The interaction effect also impacts the RFV of forage and increases during the rainy season compared with that in the dry season. The maximum RFV is observed during the rainy season across all GIs, highest in the OG site while followed by the MG and NG sites, and lowest during the dry season in the NG site, followed by the MG and OG sites. *A. kenyensis, C. aucheri* and *C. roxburghiana* show the best, the second best and the third best forage qualities respectively based on our research results.

Based on the RFV data, the highest ranking grass species has high CP and ash contents and low fiber components, indicating that high forage quality is generally related to high CP and low fiber contents. Based on this, RFV is assumed to have a direct relationship with CP and ash contents and an inverse relationship with fiber components in forages. All grass species show significantly ($P < 0.05$) higher CP, DMD, DMI, and RFV contents and lower ADF, NDF, and ADL contents in the rainy season than in the dry season. The seasonal variation is caused by maturity and age difference of forage grass species and results in variation in nutritional compositions of forage grass species within the same grazing site.
**Table 3  Interaction effects of seasonal variation and GI on forage nutritive value of each grass species.**

| Treatment | Grass species | Forage nutrient compositions (%) | | | | | | | |
|---|---|---|---|---|---|---|---|---|---|
| | | **CP** | **ADF** | **NDF** | **ADL** | **Ash** | **DMD** | **DMI** | **RFV** |
| Rs X NG | *C. roxburghiana* | 5.9[ab] | 38.1[aaB] | 53.2[aA] | 21.5[aA] | 14.1[aa] | 59.2[aaA] | 2.3[aaA] | 106[a] |
| | *C. ciliar* | 10.9[bE] | 44.7[b] | 56.3[b] | 30.8[b] | 17.7[bA] | 54.1[b] | 2.1[bdB] | 88[b] |
| | *C. aucheri* | 10.6[cC] | 34.9[cA] | 50.4[cB] | 28.8[bB] | 13.5[ca] | 61.7[cB] | 2.4[aaA] | 115[caA] |
| | *A. kenyensis* | 2.9[d] | 37.4[daE] | 47.2[dC] | 23.2[c] | 15.7[dbB] | 59.8[aaA] | 2.5[aaC] | 116[daA] |
| | *D. milanjiana* | 7.8[ebC] | 47.1[e] | 60.0[e] | 33.7[d] | 16.4[cbB] | 52.2[e] | 2.0[bdB] | 81[e] |
| Rs X MG | *C. roxburghiana* | 7.8[cC] | 34.3[ceA] | 48.1[cC] | 17.9[c] | 15.8[ccB] | 62.2[cbB] | 2.5[baC] | 121[cB] |
| | *C. ciliari* | 16.1[dD] | 41.3[db] | 50.5[dB] | 25.1[d] | 19.9[dC] | 56.7[dc] | 2.4[baA] | 105[d] |
| | *C. aucheri* | 12.8[eE] | 31.8[e] | 45.3[eA] | 21.2[eaA] | 16.5[bcB] | 64.1[e] | 2.6[baC] | 129[e] |
| | *A. kenyensis* | 4.8[c] | 35.1[ee] | 41.9[c] | 19.6[baA] | 18.0[bdD] | 61.6[bbB] | 2.9[cE] | 138[bC] |
| | *D. milanjiana* | 10.2[bA] | 41.2[bb] | 54.1[bA] | 29.1[aB] | 17.5[adA] | 56.8[ac] | 2.2[eB] | 97[a] |
| Rs X OG | *C. roxburghiana* | 9.9[eE] | 29.5[e] | 43.9[e] | 14.7[e] | 17.3[bAD] | 65.9[ee] | 2.7[eeE] | 138[eC] |
| | *C. ciliari* | 18.9[c] | 38.2[aB] | 45.2[cC] | 21.3[cA] | 21.2[afe] | 59.1[aA] | 2.7[deE] | 124[cB] |
| | *C. aucheri* | 16.7[dD] | 30.0[d] | 38.8[b] | 18.1[d] | 19.2[df] | 65.5[de] | 3.1[cb] | 157[db] |
| | *A. kenyensis* | 6.9[bcC] | 31.9[c] | 37.1[e] | 13.1[a] | 20.3[beC] | 64.0[c] | 3.2[db] | 159[ab] |
| | *D. milanjiana* | 11.2[aE] | 34.6[aA] | 49.9[aB] | 24.5[b] | 21.9[ee] | 61.9[bB] | 2.4[aA] | 115[bA] |
| Ds X NG | *C. roxburghiana* | 2.2[a] | 55.2[aa] | 72.8[aA] | 49.9[a] | 8.1[aa] | 45.9[aa] | 1.6[aaA] | 56.9[a] |
| | *C. ciliari* | 6.6[bB] | 54.4[ba] | 88.5[bb] | 53.1[e] | 12.0[bA] | 46.5[ba] | 1.4[bb] | 50.5[b] |
| | *C. aucheri,* | 4.4[cC] | 49.7[cA] | 77.0[cB] | 51.2[bb] | 9.1[cB] | 50.2[cA] | 1.6[daA] | 62.3[cA] |
| | *A. kenyensis* | 1.3[d] | 48.0[d] | 69.3[dC] | 39.4[cA] | 8.9[daB] | 51.5[a] | 1.7[caA] | 67.9[d] |
| | *D. milanjiana* | 3.5[eDC] | 59.9[e] | 87.9[eb] | 51.6[db] | 10.4[c] | 42.2[e] | 1.4[bb] | 45.8[e] |
| Ds X MG | *C. roxburghiana* | 3.9[cC] | 49.3[ccA] | 64.6[c] | 38.6[cA] | 9.3[c**B**] | 50.3[cbA] | 1.9[acB] | 74.1[ca] |
| | *C. ciliari* | 8.2[dD] | 49.9[dcA] | 77.1[dB] | 48.0[d] | 14.8[d] | 50.0[dbA] | 1.6[beA] | 62.0[dA] |
| | *C. aucheri* | 7.2[e] | 41.8[ee] | 68.8[eC] | 41.8[eB] | 11.2[bbA] | 56.3[e] | 1.7[dA] | 74.2[ea] |
| | *A. kenyensis* | 3.1[cD] | 44.2[ee] | 63.7[c] | 33.7[b] | 11.3[bbA] | 54.5[b] | 1.9[ccB] | 80.3[b] |
| | *D. milanjiana* | 5.0[bA] | 56.8[b] | 79.1[b] | 47.9[a] | 11.7[abA] | 44.7[a] | 1.5[deA] | 52.0[a] |
| Ds X OG | *C. roxburghiana* | 6.8[eB] | 38.7[ee] | 59.3[ee] | 29.7[e] | 9.9[bB] | 58.8[ee] | 2.0[adB] | 91.2[eb] |
| | *C. ciliari* | 11.4[c] | 46.5[aB] | 66.6[c] | 42.8[cB] | 14.0[a] | 52.7[a] | 1.8[ecB] | 73.5[c] |
| | *C. aucheri* | 9.1[dD] | 39.4[dde] | 59.4[be] | 37.4[dc] | 13.1[d] | 58.2[de] | 2.0[bdB] | 90.2[db] |
| | *A. kenyensis* | 4.4[bC] | 40.3[cd] | 61.1[e] | 26.6[a] | 11.9[bA] | 57.5[c] | 2.0[bdB] | 89.1[a] |
| | *D. milanjiana* | 5.7[aA] | 51.4[a] | 72.3[aA] | 37.3[bc] | 13.3[e] | 48.9[b] | 1.7[dcA] | 64.4[b] |

**Notes.**

Values in columns with different lowercase letters (a, b–etc.) are significantly different ($p < 0.05$) and values with the some second double lowercase letters under the some treatment (aa, ba, cc, bc—etc.) and values with both lowercase and uppercase letters across the treatment (aB, abA, acBD–etc.) are indicated not significant difference ($p > 0.05$). GI, grazing intensity; Ds, dry season; Rs, rainy season; NG, non- grazing; MG, moderate grazing; OG, over grazing; CP, crude protein; ADF, acid detergent fiber; NDF, neutral detergent fiber; ADL, acid detergent lignin; DMD, dry matter digestibility; DMI, dry matter intake; RFV, relative feed/forage value.

## Evaluating the proportions of nutritional contents in grass using forage quality index

Using Principal Component Analysis (PCA), the relationships between nutritional contents and various affecting factors are evaluated. The correlation matrixes of forage nutritional contents related to the impact of the seasonal variation in combination with GI and GI independently are analyzed and illustrated (Tables 4 and 5) respectively. The plotted

Table 4 Spearman's correlation coefficients of forage nutrient contents in the rainy and dry seasons at different GI.

| | RS | | | | | | | | Ds | | | | | | | |
| | CP | ADF | NDF | ADL | Ash | DMD | DMI | RFV | CP | ADF | NDF | ADL | Ash | DMD | DMI | RFV |
|---|---|---|---|---|---|---|---|---|---|---|---|---|---|---|---|---|
| CP | 1.00 | −.41 | −.46 | −.28 | .70 | .39 | .45 | .45 | 1.00 | −.37 | −.27 | .04 | .35 | .37 | .28 | .34 |
| ADF | | 1.00 | .92 | .61 | −.54 | −.99 | −.86 | −.92 | | 1.00 | .87 | .01 | −.16 | −1.0 | −.88 | -.96 |
| NDF | | | 1.00 | .64 | −.67 | −.91 | −.96 | −.97 | | | 1.00 | .04 | .01 | −.87 | −.98 | -.95 |
| ADL | | | | 1.00 | −.46 | −.60 | −.67 | −.68 | | | | 1.00 | .07 | −.01 | −.13 | -.11 |
| Ash | | | | | 1.00 | .53 | .72 | .69 | | | | | 1.00 | .16 | .06 | .11 |
| DMD | | | | | | 1.00 | .85 | .91 | | | | | | 1.00 | .88 | .96 |
| DMI | | | | | | | 1.00 | .99 | | | | | | | 1.00 | .97 |
| RFV | | | | | | | | 1.00 | | | | | | | | 1.00 |

Notes.
Rs, rainy season; Ds, dry season.

Table 5 Spearman's correlation coefficients of forage nutrient contents at different GI.

| | NG | | | | | | | | MG | | | | | | | |
| | CP | ADF | NDF | ADL | Ash | DMD | DMI | RFV | CP | ADF | NDF | ADL | Ash | DMD | DMI | RFV |
|---|---|---|---|---|---|---|---|---|---|---|---|---|---|---|---|---|
| CP | 1.00 | .52 | .72 | .70 | .96 | −.53 | −.78 | −.70 | 1.00 | .44 | .66 | .46 | .96 | −.53 | −.76 | −.63 |
| ADF | | 1.00 | .94 | .92 | .69 | −.99 | −.90 | −.96 | | 1.00 | .90 | .94 | .52 | −.95 | −.87 | −.95 |
| NDF | | | 1.00 | .99 | .86 | −.95 | −.99 | −.99 | | | 1.00 | .95 | .75 | −.91 | −.99 | −.98 |
| ADL | | | | 1.00 | .84 | −.93 | −.98 | −.99 | | | | 1.00 | .61 | −.86 | −.90 | −.95 |
| Ash | | | | | 1.00 | −.70 | −.91 | −.85 | | | | | 1.00 | −.55 | −.82 | -.71 |
| DMD | | | | | | 1.00 | .91 | −.96 | | | | | | 1.00 | .91 | .96 |
| DMI | | | | | | | 1.00 | .99 | | | | | | | 1.00 | .97 |
| RFV | | | | | | | | 1.00 | | | | | | | | 1.00 |

| | OG | | | | | | | |
| | CP | ADF | NDF | ADL | Ash | DMD | DMI | RFV |
|---|---|---|---|---|---|---|---|---|
| CP | 1.00 | .39 | .40 | .60 | .91 | −.38 | −.58 | −.53 |
| ADF | | 1.00 | .98 | .94 | .68 | −.99 | −.85 | −.94 |
| NDF | | | 1.00 | .96 | .71 | −.97 | −.93 | −.99 |
| ADL | | | | 1.00 | .83 | −.94 | −.95 | −.99 |
| Ash | | | | | 1.00 | −.66 | −.84 | −.80 |
| DMD | | | | | | 1.00 | .84 | .95 |
| DMI | | | | | | | 1.00 | .97 |
| RFV | | | | | | | | 1.00 |

Notes.
NG, non-grazing; MG, moderately grazing; OG, over grazing.

eigenvalues are obtained from the correlation matrixes and variation also calculated and explained by the components (Fig. 5).

In Tables 4 and 5, there is a strong negative correlation between CP and ash contents, and fibers (ADF, NDF, and ADL) are observed in all grass species given different grazing season, GI rate, and their interaction effect. RFV shows a strong negative correlation with ADF during the dry season and at the NG site. Component loadings with varimax rotation,
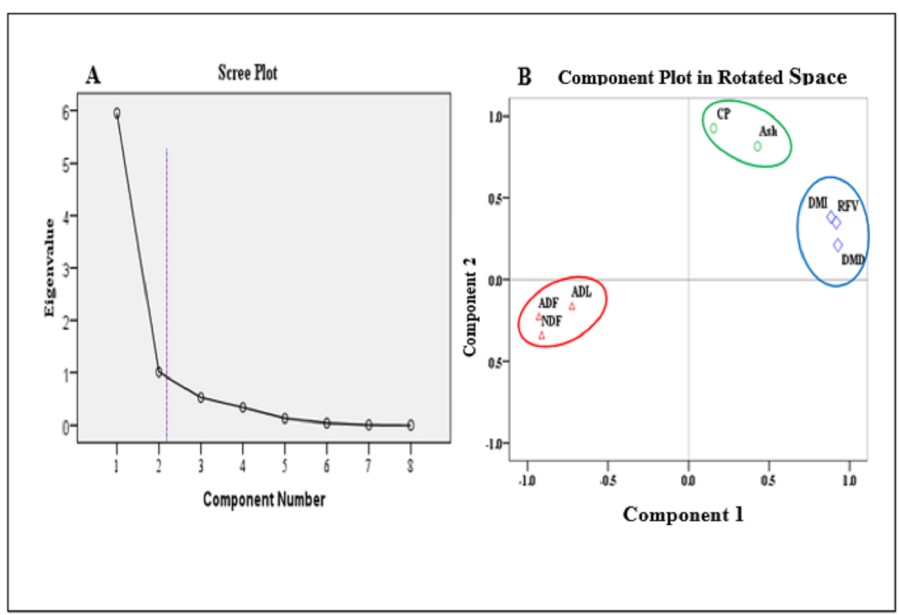

**Figure 5** Scree plot: Eigenvalues plotted in descending order (A) and Principal Components in a two-dimensiotal space (B).

**Table 6   Rotated component matrix for nutritional components of forage species data (Extraction method: Principal Component Analysis. Rotation method: Varimax with Kaiser Normalization).**

| | Total variance explained | | | | | |
|---|---|---|---|---|---|---|
| Component | Extraction sums of squared loadings | | | Rotation sums of squared loadings | | |
| | Total | % of Variance | Cumulative % | Total | % of Variance | Cumulative % |
| 1 | 5.954 | 74.427 | 74.427 | 4.925 | 61.564 | 61.564 |
| 2 | 1.011 | 12.641 | 87.067 | 2.040 | 25.503 | 87.067 |

as well as the eigenvalues show that there are only two components with eigenvalues higher than one (Fig. 5A) and the total variance is 87.067% (Table 6). Component 1 contains 60.564% of the total variance of forage nutrient contents (CP, ash, DMD, DMI, and RFV) and Component 2 contains 25.503% of the total variance of forage nutrient contents (ADL, NDF, and ADL) (Fig. 5B). On the one hand, a positive correlation exists between CP, DMD, DMI and RFV, as well as between ADF, ADF and ADL of the forage nutrient contents. On the other hand, a negative correlation between fiber contents (ADF, NDF, and ADL) and CP, ash, DMD, DMI, and RFV is observed (Fig. 5B).

# DISCUSSION

In general, our results indicate that the forage nutritive value of all those dominant grass species increases when GI rate increases, corresponding to the findings of previous studies conducted in other arid and semi-arid rangelands across the globe (*Fanselow, Schönbach*

*& Gong, 2011*; *Haiyan et al., 2016*; *Schiborra, Gierus & Wan, 2009*). Rapid increase in GI causes grazing livestock to eat young, regrown protein-rich grasses (*Gete & Gemedo, 2019*; *Mysterud, Hessen & Mobaek, 2011*). As a result, the maturation period of forage grass species is shortened, and the fiber content (ADF, NDF, and ADL) of forage is reduced, whereas the CP content increases when GI increases (*Yuan & Hou, 2015*), in direct agreement with the data recorded in our current study.

Forage maturity is inversely related to CP content and directly related to fiber content. The amount of CP content is used as an indicator of forage nutritive value, meaning that high forage quality is associated with high CP and low fiber content (*Miao et al., 2015*; Zhai et al., 2018). Typically, high CP content is inversely correlated to fiber content (Zhai et al., 2018). Based on the linear regression analysis results, the highest CP content value across all grass species appears in the OG, and the lowest in the NG. The highest fiber content across all grass species is found in the NG and the lowest in the OG. Similar results are reported by *Wang, Wang & Zhou (2011)* in a study conducted in Inner Mongolia, and also by *Miao et al. (2015)* in a study conducted on the north-east edge of Qinghai-Tibetan Plateau.

Furthermore, our conclusions are consistent with several studies conducted both at the national and international levels in arid and semi-arid rangelands. For example, it was reported that forages with high nutritive value were observed in areas with high GI (*Gemedo, 2020*; *Habtamu et al., 2013*; *Zhang, Chen & Cheng, 2015*), and the forage nutritive value was enhanced by GI. In the Teltele rangeland, forage nutrient contents show a significant difference ($p < 0.05$) across all grass species when GI varies. The grass species show a higher amount of DMD, DMI, and FRV when GI increases, and DMI is considered as a positive indicator of forage quality (*Arzani et al., 2006*).

Compared with the nutritive value of species, a higher nutritive response to GI was observed for *C. roxburghiana* and *A. kenyensis*, probably because that grazing animals found the species more acceptable at any point of time, resulting in less maturity and faster regrowth rate (*Selemani et al., 2013*; *Wan, Bai & Schönbach, 2011*). From this result, we can understand that rangeland management intensity highly affects the forage nutritive value, and grass species in the grazing site have different coping mechanisms to grazing including grazing tolerance (*Gamoun, 2014*; *Ren, Han & Schönbach, 2016*).

When collecting our sample data, the weather condition of the study site is in a normal situation (with no special climate change like drought or flooding). The results show that the forage nutritive value is higher during rainy season than in dry season, which is highly consistent with previous studies conducted by *Haiyan et al. (2016)* and *Müller, Dickhoefer & Lin (2014)*. In arid and semi-arid rangeland areas of Ethiopia, it was reported that seasonal variation has a significant influence on the nutritional quality of key forage species (*Hussain & Mufakhirah, 2009*; *Teka et al., 2012*). Our results are highly in line with the findings reported by the above authors.

In the Teltele rangeland, water scarcity is the major limiting factor for grass species growth. Higher precipitation in the rainy season increases soil water availability and improves species composition. The CP concentration is high during the rainy season because the mineralization rate and nitrogen assimilation of grass species becomes higher (Fig. 6). During the dry season, there is a scarcity of precipitation and consequently a

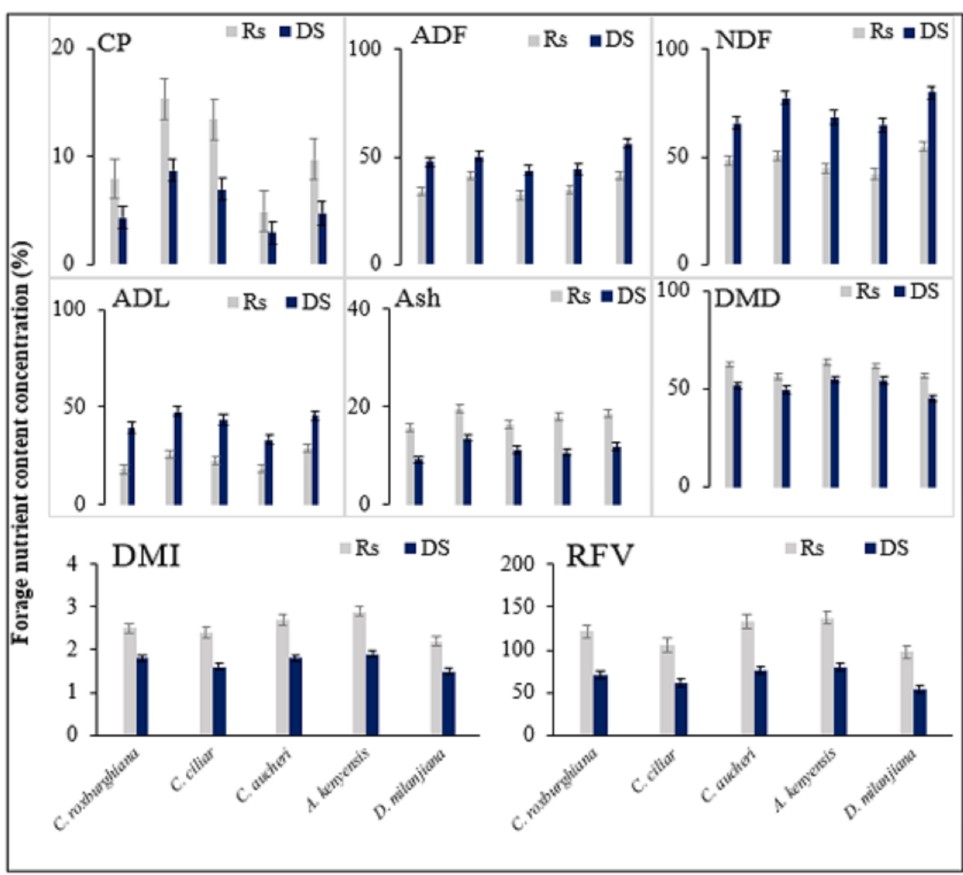

**Figure 6** **The mean forage nutrient concentration (%) at rainy season and dry season under different grazing intensity.** Error bars indicate standard error. Ds, dry season; Rs, rainy season; CP, crude protein; ADF, acid detergent fiber; NDF, neutral detergent fiber; ADL, acid detergent lignin; DMD, dry matter digestibility; DMI, dry matter intake; RFV, relative feed/forage value.

slow regrowth rate, and the forages are highly mature, resulting in high fiber and low CP concentration (*Adogla, Amaning & Ahunu, 2014*; *Gete & Gemedo, 2019*; *Shah & Paulsen, 2003*). Compared to the interactive effect of season and GI, the highest forage nutritive value is recorded at rainy season × OG (overgrazing site during the rainy season), whereas the lowest value is recorded at dry season × NG (non-grazing site during the dry season) across all grass species. And *C. roxburghiana* and *A. kenyensis* are grass species with higher forage nutritive value in both rainy and dry seasons in all GI treatments.

In our study site, grazing reduces the abundance of mature forages and accelerates the regrowth of new grass, leading to less resistance to drought and sensitivity to water loss and causing a significant variation in nutritive value. But still, there is a limitation of data on forage quality during the early growth period since the major impact on CP occurs during the early grazing period (*Rawnsley et al., 2002*; *Sollenberger, 2007*). Therefore, rangeland management practices and pastoralists should consider different grazing seasons to obtain the required amount and quality of forage for their livestock.

In general, our data indicates that rangeland grass species differ greatly in nutritive value, especially with different GI and seasonal factors. Our data is highly supported by previous studies conducted in arid and semi-arid rangelands around the globe, which revealed the complex impact of both GI and seasonal variation on the forage nutritive value. Such studies included the research conducted by *Schönbach, Wan & Gierus (2012)* in Inner Mongolia, by *Zhang, Chen & Cheng (2015)* in Qilian Mountains, *Islam, Razzaq & Shamim (2018)* in Pakistan, and by *Mountousis, Papanikolaou & Stanogias (2008)* in South Europe.

Our research results indicate that the forage quality of the dominant grass species in this study shows a significant ($P < 0.05$) difference. From the recorded data of dominant grass species, *C. roxburghiana* and *D. milanjiana* show the highest and lowest forage quality respectively across all GIs. A negative correlation is found between forage CP and RFV with fiber (ADF, NDF, and ADL) content, and a positive correlation is found between CP, DMD, DMI, and RFV for all species across all GIs and seasons. Our findings are in line with the conclusions reported by *Lin et al. (2011)*, and are in agreement with the data reporting that high Nitrogen (N) forage content has a direct linkage with good nutritional quality (*Cao et al., 2011*). The negative relationship between N (CP) and the fiber content of forage is a major indicator of rangeland forage grass species regrowth rate and maturity (*Haiyan et al., 2016*). Furthermore, the linkage between forage nutritive value indexes is highly affected by both GI and seasonal variation.

Our research results have great implications and can be used as a reference for sustainable management of arid and semi-arid rangelands in Teltele and other areas in Ethiopia, as well as other parts of the world with similar conditions. Since the forage nutritive value fluctuates due to GI and seasonal impact, pastoralists shall make appropriate preparation for the dry season and when over-degradation happens through collecting and providing different supplementary feeds for better livestock management and productivity.

The current ongoing grazing intensity and irregular seasonal change may cause rapid rangeland degradation and result in a shortage of forage for livestock grazing, which may lead to social, economic, and political instabilities in the study site and the country in general. The research finding thus plays a critical role in providing information, minimizing the risk of rangeland degradation and cutting the costs of living for both human and livestock. Contrary to our second hypothesis, GI and seasonal variation show a significant impact on the forage nutritive value of the dominant grass species in the study site. The nutritive value of the grass species in Teltele rangeland is more responsive to grazing disturbance, indicating that GI assessment in terms of forage nutritive value is highly important and scientifically recommended for sustainable rangeland management. Our first hypothesis is approved.

## CONCLUSIONS

The OG site maintains relatively higher CP and less fiber content in all grass species compared with other GI sites. Seasonal variation is also one of the major determinant factors for forage nutritive value. Higher CP and less fiber are recorded during the rainy

season than in the dry season. Besides the forage nutritive value, both GI and season significantly influence the availability and amount of forage species for grazing. And the shortage of forage under high grazing intensities reduces the livestock carrying capacity of rangeland. Moreover, the exclusion of rangeland from livestock grazing does not necessarily improve forage quality, since high CP and low fiber concentrations are linked with the GI rate. In the Teltele rangeland, desirable grass species are far from abundant, even in areas where grazing is restricted. This indicates that the most urgent action is to restore these rangelands to a state where dominant species are more prevalent. Therefore, to balance forage availability and quality based on the demand for livestock grazing, adopting sustainable rangeland management strategies like rotational grazing and maintaining grazing intensity at a moderate level are important and recommended for forage producers and pastoralists.

## ACKNOWLEDGEMENTS

Our great thanks go to the local community and stakeholder of the Teltele district for giving us the basic information that are still the challenge for them for our next research step.

### Funding

This study received financial support from CAS- TWAS fellowship program and African Great Green Wall Adaptation Technical Cooperation Research and Demonstration (2018YFE0106000), Science and Technology Partnership Program, Ministry of Science and Technology of China (Grant No. KY 201702010), and International Cooperation and Exchanges NSFC (Grant No. 41861144020). There was no additional external funding received for this study The funders had no role in study design, data collection and analysis, decision to publish, or preparation of the manuscript.

### Grant Disclosures

The following grant information was disclosed by the authors:
CAS- TWAS fellowship program and African Great Green Wall Adaptation Technical Cooperation Research and Demonstration: 2018YFE0106000.
Science and Technology Partnership Program.
Ministry of Science and Technology of China: KY 201702010.
International Cooperation and Exchanges NSFC: 41861144020.

### Competing Interests

The authors declare there are no competing interests.

### Author Contributions

- Yeneayehu Fenetahun, Yuan You, Tihunie Fentahun, Xu Xinwen and Wang Yong-dong conceived and designed the experiments, performed the experiments, analyzed the data,

prepared figures and/or tables, authored or reviewed drafts of the paper, and approved the final draft.

## Data Availability

The raw data are available in the Supplemental File.

## Supplemental Information

Supplemental information for this article can be found online at http://dx.doi.org/10.7717/peerj.12204#supplemental-information.

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
