# Peer review of "Effects of grazing intensity on forage nutritive value of dominant grass species in Borana rangelands of Southern Ethiopia"

_PeerJ, doi:10.7717/peerj.12204_

## Round 0.1 · original submission · Major Revisions

Dear Yeneayehu Fenetahun,

Thank you for your submission. Please review and respond to the comments from two external reviewers carefully. I have also gone through your submission, particularly your tables and figures. I agree with the comments from both reviewers. There are values in the study. However, I have to be very frank to you that the manuscript is quite poorly written. There are some significant issues in the experiment design, analysis, and even the lab procedure to be addressed/clarified before it can be critically reviewed again.

With these concerns, the decision would be Reject or Re-submission. So please treat this Major revision as a re-submission. A re-focus on the interactive effect of grazing intensity and dry and wet season on forage quality may be better received. Please include detailed information about the experimental design, the experimental unit, and replication and justify the statistical approaches used.

My general comments:

Research motivation and hypothesis
High grazing intensity indirectly results in higher nutritional value forage as herbage consumed by animals is less mature. This is somewhat common knowledge to range managers and ranchers. Therefore the hypothesis 1 is very weak, especially the increase in nutritional value associated with overgrazing is not something you are practically aiming at or managing for. The author may want to work hard to justify the research motivation and the true implication of this study. Hypothesis II is less apparent, and therefore it is more interesting if it can be discussed along with the primary productivity, grazing intensity, and livestock productivity.

Experiment design and analysis
The major concern is the lack of details in experimental design, making it hard to evaluate all statistical approaches used. The brief introduction of the site, plot, quadrats in the text is neither adequate nor sufficient. A graphic description such as Figure 3 is a good way to present an experimental layout, but the current form of figure 3 fails to provide enough information to serve this purpose. Please specifically address or describe the following in revision:

• Where are the NG, MG, OG sites located, how are they selected (randomly or by a grazing intensity gradient)?
• How was each site managed (cattle, sheep...) during the study?
• Any difference in climate and soil types?
• How have you handled the data from quadrants, plots, sites, and what were your true replicates (n?).
• You may revise and combine Figures 1 and 3 to reflect the experimental layout in the field, which will help evaluate whether your statistical approaches are adequate.

Reviewer 2 pointed out that your grazing intensity is a category factor, making all your linear regression meaningless. Plus, you may
have made a mistake in describing the stocking rates

As for all the above issues, I have a reasonable doubt about your results (Table 2 - 3) and the mean test. Also, the second double lower case letters are too complicated and maybe unnecessary.

Unclear, misinformation, or wrong information
• Rainfall - Please describe how did you quantify the rain? Monthly precipitation from 200 - 800 mm is not possible for semiarid rangeland. If you did not quantify rainfall, you should not present new data in your results.

• The grazing intensity or the unit of grazing intensity is unclear or wrong.

• PET - It is unclear the purpose of providing the potential evapotranspiration. Is that an annual PET? Please make sure that the range (700-3000 mm) is correct.

• Temperature for drying samples - Please review the standard procedure for drying plant samples for nutrient analysis. Reviewer 2 has a great concern about your drying temperature. "Dry directly at 105oC may change the qualitative characteristic of the grass, and that is the key point of the manuscript."


English syntax, grammar, and general writing
" the manuscript needs a thorough revision of English usage, as there are many grammar shortcomings in terms of verb tenses and syntax errors that make reading the paper difficult." I would strongly recommend the author to seek professional editing services for help.

When you have the result section and discussion section separately, please only focus on your findings (statements) in the result section. Do not include discussion or explanation in results.

Fig. 1 - Over grazing site , Un grazing site - to be consistent to the man text
Fig. 2 - Please double check the rainfall value for each month
Fig. 3 - Please consider revising and combining Fig.3 with Fig. 1 to orient the readers and help to understand the experimental design,
Fig.4 - Standard Error and n?
Fig. 5 Delete?
Fig. 7 - Redundancy with the table? n?

Reviewer 1 ·

Basic reporting

The paper is interesting, researching the changes observed in the nutritional value of semi-arid grasslands in the rainy and the dry season. However, the manuscript needs a thorough revision of English usage, as there are many grammar shortcomings in terms of verb tenses and syntax errors that make reading the paper difficult.
Line 110-111: There is a mistake on describing the stocking rates since it states that moderate grazing intensity was 6 ha/AU. In comparison, the overgrazing intensity was 12 ha/AU when this figure means less severe grazing intensity.
Authors must revise these concepts and figures
Line 130: Should be College, not collage.
Line 245 – 275: Very long paragraph difficult to read and follow. Must divide into shorter paragraphs addressing just one topic per paragraph and not more than five sentences per paragraph.
Line 277-305: Same comment, one very long paragraph. It has to be divided into shorter paragraphs.

Experimental design

Authors followed accepted methods for the evaluation of rangelands and the analyses of results.

Validity of the findings

Findings are valid, but writing has to be thoroughly improved to make the paper fluent and clear.
Authors must address the implications of their study beyond the local importance to the study region so that the manuscript is of interest and useful for an international readership.

Additional comments

It is an interesting paper on the changes observed in the nutritional value of semi-arid grasslands in the rainy and the dry season. However, the manuscript needs a thorough revision of English usage, as there are many grammar shortcomings in terms of verb tenses and syntax errors that make reading the paper difficult.
Authors use very long continuous paragraphs. The paper must be thoroughly revised before it can be accepted for publication.

Reviewer 2 ·

Basic reporting

1) Figures and tables:

Some figures and tables are presenting the same data. For example, Figure 7 and Table 3. It is necessary to review all figures and tables to decide which are necessary.

Figure 5: What is GI in axes “X”? It was not described in material and methods.


2) I couldn't found the raw data.

3) Hypotheses.
Is not clear to me why the authors building these hypotheses? In line 246 is showed some references (Fanselow et al., 2011; Haiyan et al., 2016; Schiborra et al., 2009), confirming that GI increases the forage nutritive value. I think that need a clear introduction to postulate the current hypothesis.

Experimental design

Is not clear what kind of experimental design was used. That needs to be much clearer.

Figure 1: What experimental design is that? How the repetition was randomly sorted?

Line 118 and Figure 3: What means 5 main plots? Is not clear.

Line 144: General Linear Model (GLM) is not the appropriate way to analyze the data. There were two seasons (dry and rain) and I understood that was used the same plot to collect the samples (repeated measurements), for that are necessary to use MIXED model.

Line 143: Linear regression model. To that, the quantitative factor describes in line 110 needs to be more specific. Non-grazing (NG), moderate grazing (MG) and overgrazing (OG) are more like a qualitative factor in line 110. It needs to appropriately describe, to be clear that regression analysis can be done.

Validity of the findings

1) The results are valid, but the manuscript needs to be adjusted to archive the quality necessary. But the big concern that I have is about the short experimental period (6 months). To understand all effects on grassland is necessary to investigate, at last, one year, but ideally two years.

2) Conclusion: (Line 346) With the data shown is not possible to make the conclusion, that MG should be recommended. To do that is necessary to look to forage mass and animal performance.

Additional comments

A. Introduction:

• Line 86: Hypotheses. Is not clear to me why the authors building these hypotheses? In line 246 is showed some references (Fanselow et al., 2011; Haiyan et al., 2016; Schiborra et al., 2009), confirming that GI increases the forage nutritive value. I think that need a clear introduction to postulate the current hypothesis.

B. Material and methods:

• Need much more details.

• How the carrying-capacity was kept across the seasons (dry and rainy)? What kind of animal was used? How the carrying-capacity was adjusted and which frequency?

• Line 101: The soil chemical analyses were done?

• Figure 2: If possible, add the historical data about the weather condition. That is important to understanding if the experimental period was in a normal condition for the location.

• Line 104: I think that information is not so important to the manuscript.

• Line 110: non- grazing (NG) (~ 0 ha AU-1Y-1), moderate grazing (MG) (6 ha AU-1 Y-111 1), and overgrazing (OG) (12 ha AU-1 Y-1 and above), is it correct? In this way, for example, I understand that OG has more area by the animal than in MG. I think the units should be wrong.

• Line 113: How many samples, by treatment, were collected by season (dry and rainy)? When? The samples collected need to be better explained.

• Line 118: Main plots. What was that?

• Line 123: How the dominant grass species were determined? How the relative abundances (≥ 40%) was done? Was it a subjective observation?

• Line 128: Was the sample directly dried at 105oC? That normally is done after a previous dried at 55-60oC. Dry directly at 105oC may change the qualitative characteristic of the grass, and that is the key point of the manuscript.

• Line 133 and Table 1: Dry matter intake (DMI). How is it was done? There is no detail about it.


C. Statistical analyses:
• In my point of view, the analyses need to be redone. In experimental design, I make my suggestion.

D. Results:

• To review if all 7 figures and 6 tables are necessary.
• As carrying-capacity was calculated I think the authors could present the data of forage mass too, it can be helpful to the manuscript.
• Lines 169 and 177: The authors are making discussing in the results session.
• Line 191: Biomass production. There is this information (data)?


E. Discussion:
• As a suggestion, the discussion should be done without subtopics. The use of subtopics makes the discussion less fluent. Many times the authors are telling the same discussion.
• The use of the term “palatable” (Lines 47 and 271) is not appropriated. Palatable is applied to us (humans), for the animal we can not know. The appropriate term, in this case, is acceptable (dominant grasses are acceptable by the animals).

F. Conclusion:
• Line 346: With the data shown is not possible to make that conclusion, that MG should be recommended. To do that is necessary to look to forage mass and animal performance.



G. Figures and tables:

• Some figures and tables are presenting the same data. For example, Figure 7 and Table 3. It is necessary to review all figures and tables to decide which are necessary.
• Figure 5: What is GI in axes “X”? It was not described in material and methods.

---

## Round 0.2 · Major Revisions

Dear Yeneayehu Fenetahun,

Thank you for your resubmission. It is apparent that you have put quite some effort into this resubmission, including revising the hypotheses and explaining the statistical approaches. However, most of the issues raised by the reviewers and me remain unaddressed.

Specifically, the first significant concern in the previous review is on the hypothesis. For the research hypothesis, the two primary features of a scientific hypothesis are falsifiability and testability, which are reflected in an “If…then” statement summarizing the idea and in the ability to be supported or refuted through observation and experimentation. In this regard, your hypothesis 1 did not meet both requirements. You may ask yourselves whether it is necessary to develop those hypotheses in the first place. A study with clearly defined objectives is fine.

The 2nd primary concern is the lack of details in experimental design. The resubmission has added substantial text to explain the experiment design and the statistical approaches, but many important details remain unclear. For example, what were the actual stocking rates for your grazing treatment? How have you handled the data from quadrants, plots, sites, and what were your true replicates (n?).

• Grazing treatment
How many grazing units (cattle, goats, grazing/browsing time) in each of your treatments? You should have this information, and you can easily present them using stocking rates and replaced your use of carrying capacity. You have to have this information. Otherwise, this entire study will have very little value.

• Data aggregation
You have a very complicated, nested sampling structure from plot, sub-plots, quadrats. It is essential to discuss how you have handled all those data collected at different spatial scales. The previous reviewer has raised the question on your statistics. It is your responsibility to make sure to present your statistics/analysis easily understood by readers.

• Reviewer 2 pointed out that your grazing intensity is a category factor, making all your linear regression meaningless.
You did not address nor rebut this, but include Figure 4 without any change. I agree with the reviewer, and I think this is a fundamental question regarding the true purpose of research. Without knowing the fundamental driver for the ecological process, you can develop complicated models, but they would not help but hinder our understanding of ecology. If you change your grazing intensity as stocking rate, you may be able to justify the use of stocking rate as a continuous variable and try some linear regression.

• Plus, you may have made a mistake in describing the stocking rates.
You have vigorously defended your use of “ha AU-1Y-1”. You have defined your grazing intensity based on the current carrying capacity. This makes sense, but it does cause a lot of confusion. “The treatments of sample collection involved at NG (~ 0 ha AU-1Y-1), MG (6 ha AU-1 Y-1), and OG (12 ha AU-1 Y-1 and above) grazing area based on the current carrying capacity of rangeland calculated by Fenetahun et al., (2020) and physical field observation”. I have read Fenetahun et al., (2020 and I could not found any information on carrying capacity calculation done in that paper for this study site. Consequently, I do not know how you have come up with your numbers.

Since I am not a range scientist and I did some research, and it seems to me that stocking rates are the widely used metric to describe grazing intensity. It is the number of specific kinds and classes of animals grazing or using a unit of land for a specific time period. A larger number means a high stocking rate. Stocking rate is a very important and widely used concept in rangeland management, and it is easy for readers to understand.

As this paper specifically discusses the impact of dry and wet season grazing, it is critical to have the precipitation right. Even I have explicitly asked the following question in the previous review:

• Rainfall - Please describe how did you quantify the rain? Monthly precipitation from 200 - 800 mm is not possible for semiarid rangeland. If you did not quantify rainfall, you should not present new data in your results.

I am very disappointed that you have chosen to ignore this. In your revision, you have added a sentence to explain the precipitation regime and stated – “The mean annual rainfall recorded is between 450 and 700 mm”, but you still did not do anything to address your figure 2. I did an eyeball estimate on your monthly precipitation bars from figure 2, which added about 5900 mm for annual precipitation, which is nearly ten times greater than you stated. In addition, you did not explain how you have collected those data. If you did not collect this data yourself, you should not present it as results. At a minimum, you need to provide and acknowledge the data source.

Please address the issues mentioned above point by point in your resubmission. The English language must be improved, and please contact PeerJ for editing help or provide evidence of professional editing at your revision.

One of the important editorial criteria of PeerJ is the soundness of the science. The research does not need to be cutting edge, but the data collection, analysis, and presentation need to accurate and concise. I have discussed this with the editor, and we agreed that your resubmission would not be reviewed if you fail to address or rebut all the concerns which were mainly raised in the previous review.

Reviewer 1 ·

Basic reporting

The authors have made the suggested corrections so that the paper is now accepted for publication.

Experimental design

The authors have made the suggested corrections by the reviewers on the experimental desgin and methods for analyses, so that the paper is now accepted for publication.

Validity of the findings

The findings are valid, and with the corrections, the paper has been substantially improved. It is now accepted for publications.

Additional comments

The paper is now accepted for publication.

Reviewer 2 ·

Basic reporting

Point 1: Description of treatments - The way how the authors are describing the grazing intensity (GI) treatments is confusing, in the first revision both reviewers had the same difficulty to understanding. After reading the authors' answers, I still had difficulty understanding, but I guess I could understand. In my point of view, that will confuse to the readers too. What I understood was that the author did not apply the treatments, they select areas that were no-grazed (NG) or had moderated (MG) and overgrazed (OG) and calculated the carrying capacity of those areas (treatments). To clarify that, in the manuscript, will be helpful to better understanding.

Point 2: Conclusion (Line 381) – Looks like that the authors made a discussion in the conclusion section, the discussion is good but should be added in the discussion section and the conclusion should be more concise.

Point 3: Line 45 - The use of the term “palatable” is not appropriated, for an animal in the pasture is better to use “acceptable”. I made that comment in the first review.

Point 4: Line 127 – Two or three treatments?

Point 5: Line 174 - The dry matter intake (DMI) was not evaluated, was done an estimate of the potential of dry matter intake. I guess should be used as “potential dry matter intake” to avoid misunderstanding.

Point 6: Line 173 – There are no units to variables forage nutritive value as, for example, crude protein.

Experimental design

Ok

Validity of the findings

Ok

---

## Round 0.3 · Minor Revisions

It is apparent that the manuscript has improved after two rounds of revisions. However, the current version is still full of grammatical and syntax errors. Here I just point out a few grammatical and syntax errors in the first two paragraphs of the introduction to justify my statement.

L61. Although, based on the density of acceptable forage species, does not estimate the nutrition quality of forage in the grazing area (Samuels et al., 2015). No subject!

L75 - Seasonal changes in forage nutritional value shows a linear decrease. “changes ….shows”!

L78-79 – since digestibility become decreased due to high temperature resulted in a decline in the leaf-to-stem ratio (Kirch et al., 2003). Syntax error!

L81 - In Teltele rangeland, the local pastoralists select communal grazing areas for their livestock grazing for continuously- "for continuously”?

I have made it clear in the previous decision letter of my concerns about English. Apparently, my concerns were completely ignored. Please contact the PeerJ office and ask whether they can provide you technical editing to meet the journal requirement. I request evidence of professional editing at your next revision or your revision will be rejected without further review.

---

## Round 0.4 · Minor Revisions

PeerJ has made it clear that a "poorly written manuscript will be returned to the author" in its editorial criteria (https://peerj.com/about/editorial-criteria/). Although you have corrected many grammatic errors, it is my assessment that the English still does not meet the requirements or standard of PeerJ publication. In addition, the manuscript in its current form does not strictly follow the journal instruction (https://peerj.com/about/author-instructions/).

PeerJ is an open-access journal and it charges fees for publications and editing services. You may also find a colleague or friend who is a native English speaker to help. Alternatively, you may also consider withdrawing this submission and re-submitting it to a conventional journal that provides free technical editing for accepted manuscripts.

I have spent many hours handling your submissions. I hope I can help more. However, as an academic editor, I do not have the time nor the credential to perform formal English editing for publication.

---

## Round 0.5 · accepted · Accept

The technical editing has greatly improved the clarity and reliability of this manuscript. Thank you for your patience and congratulations!